# Soft Robotic Glove with Sensing and Force Feedback for Rehabilitation in Virtual Reality

**DOI:** 10.3390/biomimetics8010083

**Published:** 2023-02-15

**Authors:** Fengguan Li, Jiahong Chen, Guanpeng Ye, Siwei Dong, Zishu Gao, Yitong Zhou

**Affiliations:** Shien-Ming Wu School of Intelligent Engineering, South China University of Technology, Guangzhou 510641, China

**Keywords:** soft robotic glove, rehabilitation robotics, soft wearable robotics, hand rehabilitation, haptic device, data glove

## Abstract

Many diseases, such as stroke, arthritis, and spinal cord injury, can cause severe hand
impairment. Treatment options for these patients are limited by expensive hand rehabilitation devices
and dull treatment procedures. In this study, we present an inexpensive soft robotic glove for hand
rehabilitation in virtual reality (VR). Fifteen inertial measurement units are placed on the glove for
finger motion tracking, and a motor—tendon actuation system is mounted onto the arm and exerts
forces on fingertips via finger-anchoring points, providing force feedback to fingers so that the users
can feel the force of a virtual object. A static threshold correction and complementary filter are used
to calculate the finger attitude angles, hence computing the postures of five fingers simultaneously.
Both static and dynamic tests are performed to validate the accuracy of the finger-motion-tracking
algorithm. A field-oriented-control-based angular closed-loop torque control algorithm is adopted to
control the force applied to the fingers. It is found that each motor can provide a maximum force
of 3.14 N within the tested current limit. Finally, we present an application of the haptic glove in
a Unity-based VR interface to provide the operator with haptic feedback while squeezing a soft
virtual ball.

## 1. Introduction

The human hand, a multi-joint complex, is important to the human ability to perform activities of daily living [1]. Unfortunately, many disorders, such as stroke, arthritis, and spinal cord injury [2], can cause severe impairment of hand function. These neuromuscular disorders and traumatic events that impair motor functions can significantly deteriorate the quality of patient life [3]. Patients with functional hand impairments are required to undergo physical hand rehabilitation therapy, which mainly involves functional hand assessment and training.

The assessment of hand function relies on hand motion capture devices, which are mainly classified as non-contact and wearable. Cameras are primarily used in non-contact methods for motion capture and image analysis [4,5,6]. However, this method is highly limited by camera placement and ambient brightness and is easily obscured by the rest of the body. Currently, research on wearable methods is focused on data gloves [7,8,9,10,11]. Data gloves are efficient tools for hand motion tracking and assessing hand function among the many hand rehabilitation systems. A variety of sensors have been investigated for data gloves, such as flexible bending sensors, inertial measurement units (IMUs), etc. Flexible bending sensors are resistive sensors with the advantage of being lightweight and inexpensive [11]. However, such sensors can only provide relative position between finger segments and no information on finger position in space [12], which is crucial in virtual reality (VR) scenarios. IMUs are attractive for data gloves since they are small in size, lightweight, and highly accurate, as well as since they collect various information such as acceleration and angular velocity. A number of IMU-based data gloves have been developed. Kortier et al. proposed an ambulatory system using inertial sensors that can be placed on the hand [8]. Compared to previous static measurements, they set up dynamic tests to better match daily use. Connolly et al. proposed an IMU sensor-based electronic goniometric glove with a two-layered glove structure [9]. However, this glove has IMUs embedded in the mid-glove layer and lacks a detachable modular design. Lin et al. proposed an inertial-sensor-based data glove for hand function evaluation with a modular design [10]. But the static and dynamic tests of this glove only involve a single IMU on one finger, which does not meet the real-world requirements of simultaneous measurement and computation of multiple IMUs on an integrated glove.

Practicing the same training program repetitively may reduce the patients’ training willingness and efficiency. Interactive force feedback can be customized and programmed in VR, which can add zest to the rehabilitation program. So far, several studies have applied VR systems to the field of hand and upper limb rehabilitation. Ziherl et al. combined a 3-degree-of-freedom rehabilitation robot and a dynamic virtual environment platform to train patients’ upper limb motor skills during the process of picking and displacing [13]. Gu et al. created the Dexmo mechanical exoskeleton force feedback system [14]. However, these devices can be too bulky to allow the patient to move freely.

Recently, soft robotic gloves [15,16,17,18] have become a promising solution to overcome the limitations caused by the hard exoskeletons, which are lightweight and comfortable to wear. Soft robotic gloves with force feedback are typically powered by two sources: pneumatic pressure or tendon. Pneumatically powered gloves use fluidic elastomer actuators mounted on the glove’s finger portion [19,20]. It is simple to control and evenly distribute the pneumatic pressure over the finger area. However, pneumatic-driven gloves are heavier and bulkier in both gloves and power sources. When using tendon-driven gloves, the actuators can be placed apart from the glove’s body to prevent the oversized glove from interfering with human hand movement. Previously, researchers combined tendons with a variety of actuators, such as shape memory alloys [21], twisted string actuators [22], and DC motors [23]. The use of a motor-tendon system avoids occupying too much volume on the back of the hand, provides the sufficient force, and allows simple control. However, the current motor-tendon-driven gloves lack knowledge of whole-hand finger postures, limiting their application in assessing finger motions and rehabilitation. In addition, the general tendon-driven gloves either provide force feedback only to the thumb and index finger [21,23] or drive the five fingers using two tendons [22], exhibiting an oversimplified design and a low degree of freedom.

In this study, we present a soft robotic glove with whole-hand finger motion tracking and motor-tendon-driven force feedback for rehabilitation in VR. We first develop a data glove with fifteen IMUs for finger motion tracking using a detachable modular design. The detachable modular design enables replacing worn or broken units easily, allowing fast maintenance. A static threshold correction method and complementary filter are used to facilitate whole-hand finger motion-tracking. Both static and dynamic tests are conducted to evaluate the tracking error. In addition, a motor-tendon force feedback system is developed using five brushless motors to apply force to the five fingertips respectively when a virtual object is detected in VR. Tests are conducted on the motor to identify the relationship between the current limit and the applied force. Then, a FOC-based (field-oriented control) angular closed-loop torque control algorithm is adopted to control the motor output torque. A virtual reality interface is built using Unity, where a virtual hand is projected from our physical glove, and a virtual ball is generated and can be grasped and squeezed. The whole hardware cost is around 220 USD, which makes the glove accessible to a larger population. Our proposed glove could be used as a simple method for assessing and training hands, as well as a foundation for future applications involving soft wearable haptics, such as gaming.

## 2. Hardware Design

### 2.1. System Overview

Figure 1a shows the system architecture of the proposed system. The proposed soft robotic glove (Figure 1b) includes two main hardware subsystems: finger motion tracking for hand simulation, and force feedback. The fingers’ postures are calculated based on the attitude angle calculation algorithm and raw data collected by the data glove with fifteen IMUs. Meanwhile, a virtual hand is created in the VR scene using Unity and is simulated in real time with the calculated physical finger postures. When a user’s hand is detected grasping a virtual object (Figure 1c), the torque of the motors is controlled to provide pulling force to each finger, allowing users to feel the force of a virtual object.

The hardware for finger motion tracking consists of two main parts: (1) flexible printed circuits (FPC) and fifteen IMUs, and (2) an adapter board and microcontroller unit (MCU). The hardware for the force feedback module includes (1) five drive motors, motor encoders, and a holder; (2) a FOC drive board and MCU. Fifteen IMUs are used in conjunction with the serial peripheral interface (SPI) to transmit raw data to the MCU1, which serves for finger motion tracking, through an adapter board. The MCU1 then calculates and transmits the fingers’ postures to a personal computer (PC) via a universal serial bus (USB) and interacts with the VR scene. When the simulated hand in the VR scene grasps the object, the PC will send the touch signal to the MCU2, which serves for force feedback control, via wireless fidelity (Wi-Fi). MCU2 interacts with the FOC driver board via an inter-integrated circuit (I2C) bus and then controls the motors to apply real-time forces to the glove. In addition, MCU2 and the PC are connected to the same Wi-Fi for convenient wireless communication.

### 2.2. Hardware for Hand Simulation Based on Finger Motion Tracking

The proposed soft robotic glove includes fifteen 6-axis IMUs (LSM6DS3, ST Microelectronics) to collect data. The IMU contains a 3-axis accelerometer and a 3-axis gyroscope. To enhance wearability and improve data transmission efficiency, the proposed glove adopts an FPC design, as shown in Figure 2a. The designed FPC is 20 cm in length. It carries three pairs of 2.54-mm pin headers with a pitch of 45 mm and 50 mm (from left to right) to connect the IMU and FPC. Different from the traditional way of soldering IMUs directly [24,25], this method facilitates easy assembly and disassembly of IMU modules, allowing replacement of the worn IMU modules individually, reducing cost, and improving the life of the glove. In addition, the fixed-height pin headers help maintain the position of IMUs, improving data reading accuracy. The tail end of FPC is in golden finger type, which facilitates its connection with the flip-up FPC connector on the adapter board, saving space and assembly time.

Figure 2b depicts the placement of the fifteen IMUs on the glove. The prototype of the data glove integrated with the finger-tracking hardware is shown in Figure 2c. Cloth finger rings are sewn on each knuckle. The FPC is then taped to the corresponding finger rings to hold the IMUs. The adapter board is glued to the overhead layer on the glove and connected to an Arduino Mega2560 via jumper wires. We use SPI to transmit the raw data to the MCU. Compared to the I2C communication protocol adopted by most IMU modules [26,27,28], SPI is better suited for multi-IMU systems, which greatly reduces the total transmitting time for the fifteen IMUs. Hence, enhancing the data glove’s performance for whole-hand finger tracking in real time.

### 2.3. Hardware for Force Feedback

The force feedback part of the soft robotic glove includes five brushless motors (Gimbal 2208) for actuation, five AS5600 encoders for data conversion, and a FOC driver board (Arduino SimpleFOCShield v1, SimpleFOCProject [29]) and ESP32 (ESPRESSIF) for control. Figure 3a shows the 3D rendering of the force feedback components. A motor support assembly structure is designed to improve the integration of the device, including a shell, a bobbin, and an encoder installation unit, which are all 3D printed. The motor is mounted above the bobbin to directly drive the bobbin, which in turn controls the nylon wire wrapped around the bobbin to wind and unwind. The encoder installation unit is placed on top of the motor and is fixed inside the shell along with the motor. The motor assembly is approximately cylindrical, with a diameter of 37 mm and a height of 45 mm.

The schematic layout of the feedback wires on the back of the glove is illustrated in the upper-right of Figure 3a. One end of a feedback wire is fixed to a 3D-printed fingertip sleeve. The wire then passes through the nylon tubes inside finger rings, which serve as anchor points. Note that the wire separates into two along the finger for better force balance. Two nylon tubes are sewn onto each finger ring, which is then glued on the glove. Then, the other end of each feedback wire is wound on a bobbin. The five feedback wires are arranged in a cross pattern to ensure smooth driving. At the point of the cross, they are on top of each other without entanglement. There can be slight friction when they move against each other, but it is too small to perceive. Finally, the five motor assemblies corresponding to the five feedback wires are attached to a 3D-printed plate using Velcro for easy removal and replacement. The prototype of the integrated soft robotic glove is shown in Figure 3b. The weight of the hand-wearing part is around 50 g, and the total weight of the prototype is 450 g.

## 3. Software and Control

### 3.1. Execution Flowchart of Software Program

Figure 4 shows the execution flowchart of the software program for the VR hand training system. When a game of grasping is started in the VR system, the sensor calibration in the robotic glove is triggered. Then, the glove records the IMU data in real time, and the static threshold correction and complementary filter are used for attitude angle calculation. Next, the finger postures are solved and transmitted to Unity to simulate the virtual hand in real time. If the virtual hand touches a virtual object, a touch signal is excited, and the current limit of the drive motor is then adjusted, which tailors the output torque according to the FOC-based angle closed-loop torque control algorithm to achieve the force feedback. If no touch occurs, the output torque is maintained at a small value to keep the feedback wire tight.

### 3.2. Attitude Angle Calculation Algorithm

#### 3.2.1. Sensor Calibration

Each IMU sensor is unique and slightly different from others due to manufacturing errors. Hence, IMU sensor calibration is necessary to ensure that accurate finger joint angles can be calculated. In this study, an IMU is placed stationary on a horizontal surface to correct for zero shift. We collect 600 raw data points for calibration. For accelerometers, their angular values can be calculated by attitude solving [30]. Therefore, accelerometers are more suitable for calculating the angular offset in each direction individually. The calculation of the angles is based on Euler angles, and the rotation order is *z*-*y*-*x*, i.e., the IMU coordinate system coincides with the geodesic coordinate system at the initial moment and then rotates around its own *z*, *y*, and *x* axes in turn. The attitude angles are computed as
(1)$$\theta_{Acc_{x}}=\arctan\left(\frac{Acc_{y}}{Acc_{z}}\right),$$
(2)$$\theta_{Acc_{y}}=-\arctan\left(\frac{Acc_{x}}{\sqrt{Acc_{y}^{2}+Acc_{z}^{2}}}\right),$$
where the subscript Acc stands for the accelerometer. We average the 600 measurements of the raw data to obtain the offset angle of the accelerometer $\theta_{Acc_{offset}}$.

Because the gyroscope needs to be integrated to obtain the angle, the offset can be continuously amplified during the integration process, reducing the angle accuracy [31]. Therefore, the ${\dot{\theta }}_{Gyro_{offset}}$ is obtained by the average of 600 raw gyroscope measurements. Then, the corrected angle and angular velocity are obtained as
(3)$$\theta_{Acc_{corrected}}=\theta_{Acc}-\theta_{Acc_{offset}},$$
(4)$${\dot{\theta }}_{Gyro_{corrected}}={\dot{\theta }}_{Gyro}-{\dot{\theta }}_{Gyro_{offset}},$$
which are then input to the complementary filter to solve for the attitude angle θ.

#### 3.2.2. Static Threshold Correction

Compared to a single-IMU system, a multi-IMU system needs to read and process data from multiple IMUs, which takes more time and accumulates more errors. The IMUs may continue to perform integration even in the static state, resulting in a larger integration accumulation error. To address this issue, we propose a static threshold correction method. First, fifteen IMUs on the data glove are turned on and kept stationary simultaneously for five minutes, and the attitude angle of a random IMU is recorded during this time interval. Then, the angle of this IMU at different time steps is found to concentrate in (−0.020°, 0.025°). To improve the static-angle-solving stability, when the computed gyroscope angle is in this interval, the IMU is considered stationary, and no angle accumulation is performed.

#### 3.2.3. Complementary Filter

Sensor data fusion is necessary because dynamic motion causes accelerometer measurement perturbations and the integration process of discrete gyroscope data accumulates errors. Multiple sensor fusion algorithms have been proven effective and widely used in kinematics-related fields, such as the Kalman filter [32] and the Madgwick algorithm [33,34]. However, these algorithms are computationally expensive and time-consuming, which increases the integration accumulation error and does not meet the real-time requirements of our VR scenario applications. Therefore, this study uses a complementary filter, which is more efficient.

Figure 5 shows the flowchart of the complementary filter. Because the accelerometer has significant low-frequency characteristics, the angle $\theta_{Acc}$ calculated from the accelerometer is passed through a low-pass filter to preserve the low-frequency character. The gyroscope, on the other hand, exhibits high-frequency characteristics, so the angle $\theta_{Gyro}$ derived from the integration is passed through a high-pass filter to maintain the high-frequency character. Then, the two are multiplied by scaling factors and added as
(5)$$\theta =f\left[\theta_{prev}+{\dot{\theta }}_{Gyro_{corrected}}\cdot \Delta t\right]+\left(1-f\right)\theta_{Acc_{corrected}},$$
where θ is the final estimated attitude angle, $\theta_{prev}$ is the angle at the previous time step. Here, *f* is the scaling factor, which in this study is set to 0.96 for better-estimating accuracy. When the estimated attitude angle is obtained, the angle of each finger joint can be calculated.

Figure 6 illustrates the orientation of three IMUs on an FPC for the thumb and index finger, and interphalangeal joint angles. The IMUs have a right-handed system and are placed on the knuckles, and their positive *x* axis coincides with the finger’s longitudinal direction. For the thumb, the angle of the PIP joint is derived from the difference between IMU2 and IMU3, and the angle of the DIP joint is derived from the difference between IMU1 and IMU2. The index finger is representative of the remaining fingers other than the thumb. So, for the remaining four fingers, the angle of the MCP joint is the angle calculated by IMU3 starting from the finger’s straight state; and the angles of the PIP and DIP joints are calculated in the same way as that of the thumb.

### 3.3. Force Feedback Control Algorithm

In order to complete the force feedback in VR scenes, we need to control the torque of the drive motor. Field-oriented control (FOC) is a promising method for efficient control of brushless DC motors and permanent magnet synchronous motors, which can precisely control the magnitude and direction of the magnetic field [29]. The force feedback control algorithm in this study uses a FOC-based angular closed-loop torque control method. Compared to the approximate estimates controlled using voltage or DC current methods [35], this method enables accurate and smooth control of the torque at arbitrary rotating speeds.

Figure 7 depicts the fundamental steps of FOC. First, the acquired three-phase currents $i_{a}$, $i_{b}$, $i_{c}$ are transformed into two-phase $i_{\alpha }$, $i_{\beta }$ using Clark transform. The motor is then transformed by Park transform from a two-phase stationary coordinate system to a coordinate system that rotates with the rotor (*d* and *q* axes), and the resulting $i_{d}$ and $i_{q}$ are fed into the PID controller to produce output voltages $u_{d}$ and $u_{q}$. Finally, the AC waveform is generated by inverse Park transform and space vector modulation (SVM) to obtain $u_{a}$, $u_{b}$, and $u_{c}$ to control the motor.

Figure 8a illustrates the flowchart of FOC-based torque control. Only $i_{q}$ facilitates torque generation because the *d* axis coincides with the direction of the magnetic field inside the rotor and the *q* axis is perpendicular to it. When using FOC to control the torque, the PID controller will keep $i_{q}$ equal to the desired current $I_{desired}$ while $i_{d}$ equal to zero to maximize the torque.

Figure 8b shows the flowchart of FOC-based angular closed-loop torque control. The current angle read by the sensor in the motor is low-pass filtered and deducted from the desired angle. The result is then fed into the PID controller, and the desired speed is obtained by speed limiting. The difference between the desired speed and the processed actual speed is fed into another PID controller, and the desired current $I_{desired}$ is obtained by current limiting. Finally, $I_{desired}$ is input into the FOC-based torque control loop, which controls the motor to generate the desired torque. Based on the force feedback control algorithm, by setting the current limit, the output torque can eventually be controlled.

## 4. Experimental Validation and Application

### 4.1. Experimental Validation of Computing Static Attitude Angles

To validate the effectiveness of our IMUs and finger joint angle computation algorithm, in this section, a commercial IMU device (Wit-Motion WT9011DCL), whose brand has been used widely by multiple studies [36,37,38], is used as a benchmark for our angular tests. Its results are compared to our own IMU measurements on the proposed soft robotic glove. The WT9011DCL is placed firmly on our IMU that is to be measured, moving simultaneously with it, as shown in Figure 9. This ensures that both IMUs are in the same state at all times, and therefore the measured angles are comparable.

To validate the ability to compute the static attitude angles, a static test is first conducted. Here, we test three states of the glove: open, semi-closed, and closed states to meet the real-world use cases, as shown in Figure 9. The static tests are performed such that the finger joint’s IMU is kept stationary for fifteen minutes in each state. The attitude angles from both the commercial and our IMU are recorded during this time interval. Because the motion for the DIP joint of the index finger is relatively large, which is likely to accumulate larger errors. The static test is only conducted for the IMU1 of the index finger, which is rather representative. Table 1 lists the mean absolute error (MAE) for the three states during fifteen minutes, which is all less than 3°, indicating that our static threshold correction method is effective for keeping the accumulation error low and the attitude angle computation is efficient in static conditions.

### 4.2. Experimental Validation of Computing Dynamic Attitude Angles

Most of the previous dynamic tests only involve a single IMU or a single finger [10,39,40], and their experiments cannot directly prove the accuracy of the multi-IMU calculation. In contrast, our study turns on fifteen IMUs on the glove simultaneously and takes the time used for the calculation of all IMUs as the time interval for gyroscope integration. This test setup reflects the real-world application of the data glove and ensures that the correct attitude angle value is obtained.

The experimental equipment for dynamic tests is the same as that for static tests. Since the joint composition of the thumb is different from that of the remaining four fingers, dynamic validation is conducted on both the thumb and index finger. The accuracy of the index finger can also be used to represent the other three fingers that move in a similar manner. Starting from the fully open state, the hand bends slowly to the semi-closed and closed states, and then slowly straightens back to the initial state. The above procedure is repeated six times for testing each finger. Figure 10 depicts the dynamic test results. It is observed that our dynamic attitude angles track the commercial ones very well, indicating the effectiveness of our attitude angle algorithm under dynamic conditions. Table 2 lists the calculated mean error under dynamic tests of the six IMUs. The mean errors are small, indicating that our data glove performs fairly well in terms of angle accuracy under dynamic conditions.

### 4.3. Test of Feedback Force versus Current Limit

This study adopts a FOC-based angular closed-loop torque control algorithm, which controls the motor output torque by tailoring the input current limit. So, we set up an experiment to determine the relationship between the current limit and torque, as shown in Figure 11.

A force gauge is used to pull the wire wound in the motor assembly. The motor and the force gauge are fixed on the table to keep the feedback wire in a just-tight condition. The force gauge is fixed in line with the wire to keep the wire horizontal. Because the motor action radius is fixed, our study varies the current limit and measures the force. For each current limit value, we set the initial position of the motor such that the wire is a little slack. Then, the motor is driven to rotate, which tightens the wire, preventing the motor from rotating further. At this point, the force gauge records the blocking force of the motor, so we can measure how much feedback force a motor can provide under a specific current limit. During the test, we start from 0.05 A with an increment of 0.05 A and a maximum value of 0.55 A.

Figure 12 shows the test results for the feedback force as a function of the current limit. It is observed that the feedback force increases with the current limit. In addition, the slope of the curve increases gradually with the current limit. It is found that a single motor can generate a maximum force of 3.14 N within the range of the tested current limit. In fact, if the current limit is increased further, the feedback force will continue to rise, but 3.14 N is sufficient for humans to perceive [41]. From this relationship, one can easily adjust the feedback force by tailoring the current limit, hence the training intensity.

### 4.4. Real-Time VR Scene Application

This section demonstrates the application of our VR rehabilitation system by squeezing a virtual deformable ball in a home-built VR environment. The procedure follows our proposed automatic software program in Figure 4. We built a VR interactive scene using Unity, which includes a virtual hand and a deformable ball, as shown in Figure 13. Both the object and the fingertip of the virtual hand are assigned a certain volume of the touch area. The Appendix A shows this demonstration.

For hand simulation, first, the system performs sensor calibration. Then, the IMU system transmits the data read in real time to MCU1, and MCU1 performs static threshold correction and complementary filter on the IMU data to obtain all finger joint angles, which are then transmitted to Unity on the PC side through the serial port. Next, Unity updates the virtual hand in real time using the physical finger joint angles.

For force feedback, when the object is not detected, the current limit is kept at 0.05 A, then the motor outputs a small torque to the feedback wire, keeping it tight with slight resistance. At this time, the virtual ball is intact. When the object is detected to intersect with the touch area of the object by Unity, it sends a touch signal to MCU2 through WiFi, which triggers the force feedback system. The current limit is then increased to 0.15 A and the motor outputs an evidently larger torque to pull the feedback wire, so the user perceives a pseudo-real grip. When the user overcomes this feedback force, they pull the feedback wire further, and the virtual ball is squeezed (Figure 13). When the fingers are detected to have separate from the touch area of the object, Unity sends an exit signal to MCU2, which resets the current limit to 0.05 A, recovering the initial small torque. The current limit can be customized for different training programs and patients with various levels of motor impairment.

## 5. Conclusions

This study proposed a soft robotic glove with IMU sensing and force haptic feedback for hand rehabilitation in virtual reality. The system is lightweight (450 g), low-cost (220 USD), has high sensing accuracy, is capable of producing sufficient force, and could deliver a perception of virtual grasping. Patients with hand motor impairments may benefit from our virtual resistance training system.

The paper detailed the construction of the data glove, the force feedback module, as well as system sensing and control algorithms. We also conducted a complete validation of the data glove and motor assembly. The static and dynamic test results showed that our proposed data glove exhibits good accuracy and stability to perform finger motion tracking. The mean absolute errors in the 15-min static tests are 0.3243°, 1.1090°, and 2.6092° for the open, half-closed, and closed states, respectively. The mean error in dynamic tests is within ±3° for the thumb and ±2° for the index finger. These performance is comparable to most studies where only a single IMU or a single finger was tested, indicating that our static threshold correction method is effective for keeping the accumulation error low under static conditions, and the attitude angle computation algorithm is efficient under both static and dynamic conditions. The test of the motor assembly revealed that the motor can provide sufficient force and torque increases with the current limit. The demonstration of squeezing a virtual deformable ball proved the effectiveness of our system.

Further efforts include optimization of the number and distribution of sensors to obtain whole-hand motion and a lightweight design such as using smaller motors and an optimized motor assembly design. In addition, algorithms for automatic assessment of the motor impairment level will be developed in conjunction with a rehabilitation scale, improved VR scenes, and artificial intelligence. Furthermore, clinical tests will be conducted to validate the training system and facilitate the rehabilitation of patients.

## Figures and Tables

**Figure 1 biomimetics-08-00083-f001:**
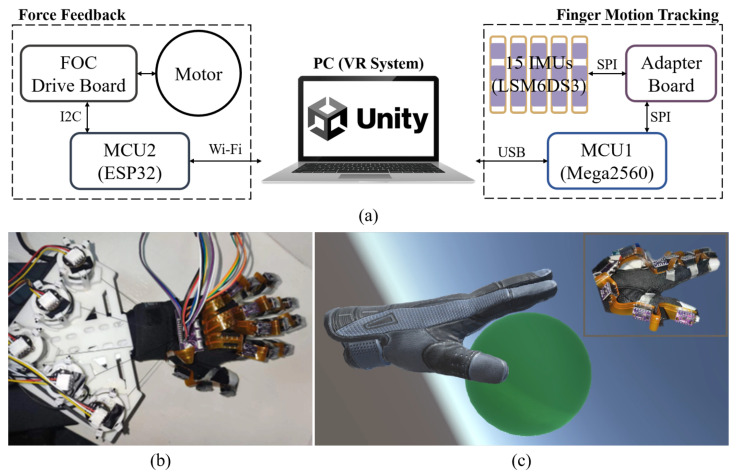
(**a**) Architecture of the proposed soft robotic glove system for rehabilitation. (**b**) Prototype of the integrated soft robotic glove. (**c**) Demonstration of grasping a virtual ball in a home-built VR scene.

**Figure 2 biomimetics-08-00083-f002:**
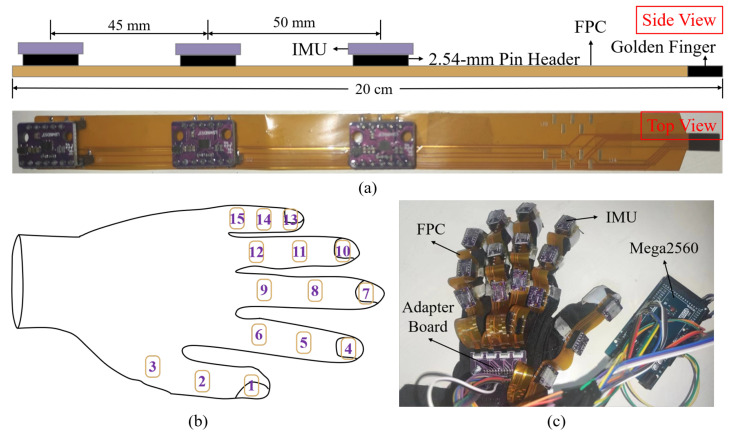
(**a**) Side view of the design drawing and top view of the prototype for an integrated FPC with three IMUs. (**b**) Placement of the IMUs on the glove. (**c**) Prototype of the data glove integrated with finger tracking hardware.

**Figure 3 biomimetics-08-00083-f003:**
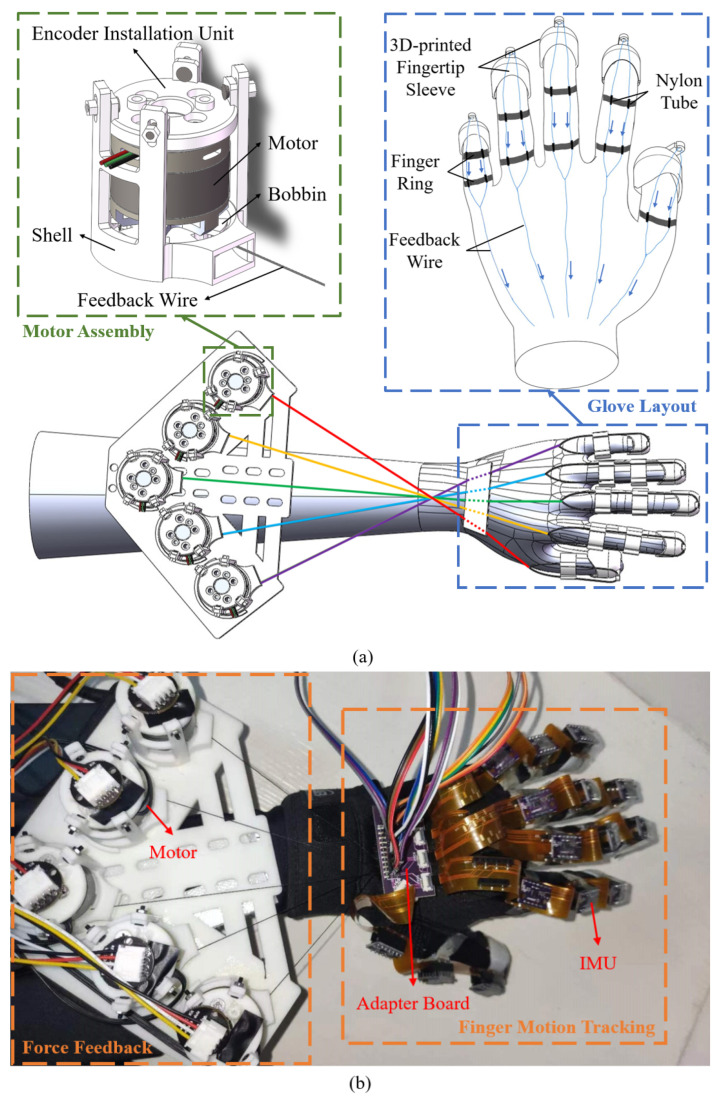
(**a**) 3D rendering of the force feedback assembly including motor assembly, glove layout, and component integration. (**b**) Prototype of the integrated soft robotic glove.

**Figure 4 biomimetics-08-00083-f004:**
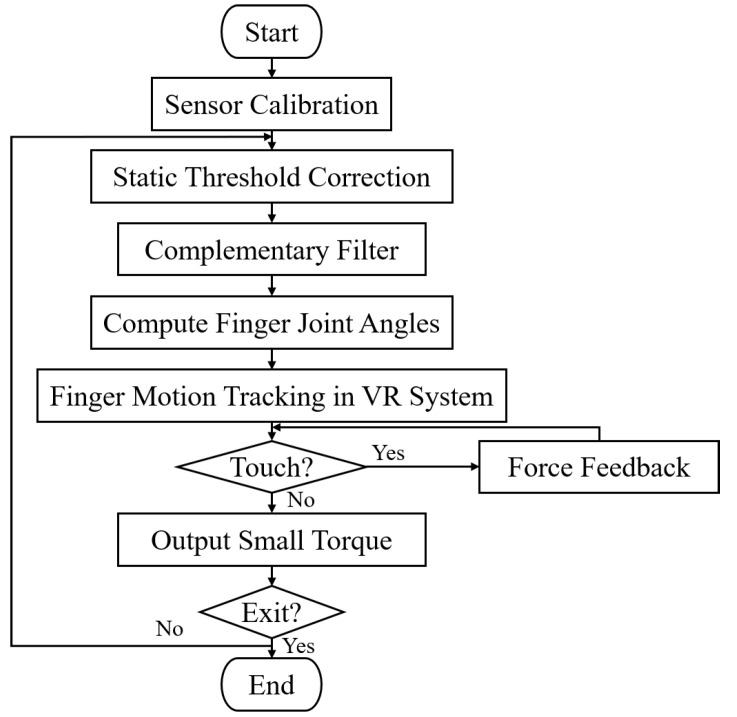
Execution flowchart of software program for the VR hand training system.

**Figure 5 biomimetics-08-00083-f005:**
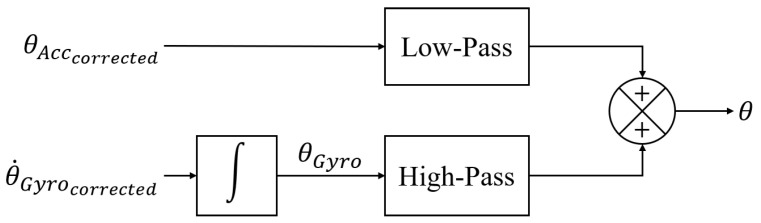
Flowchart of complementary filter.

**Figure 6 biomimetics-08-00083-f006:**
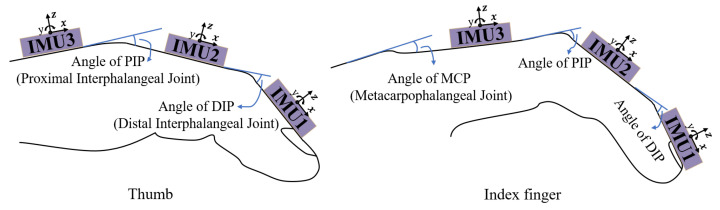
IMU orientation and angles of finger interphalangeal joints for the thumb and index finger.

**Figure 7 biomimetics-08-00083-f007:**
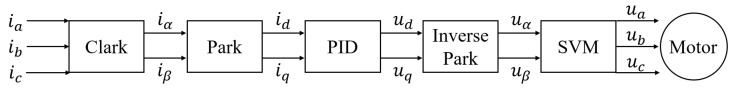
Fundamental steps of field-oriented control (FOC).

**Figure 8 biomimetics-08-00083-f008:**
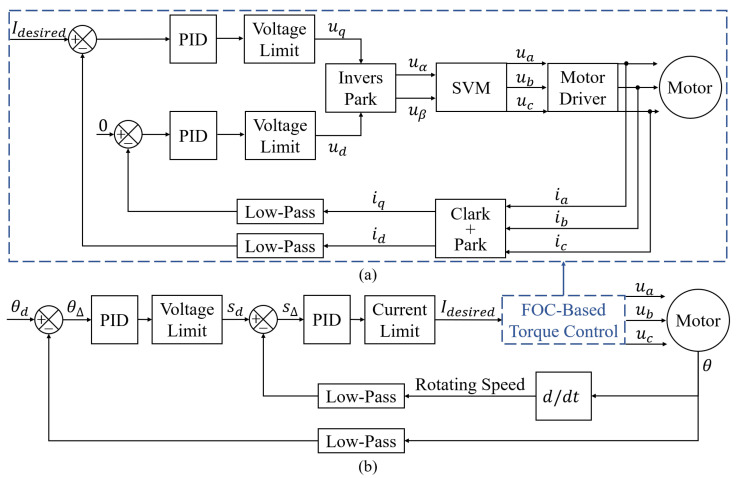
(**a**) Flowchart of FOC-based torque control. (**b**) Flowchart of FOC-based angular closed-loop torque control.

**Figure 9 biomimetics-08-00083-f009:**
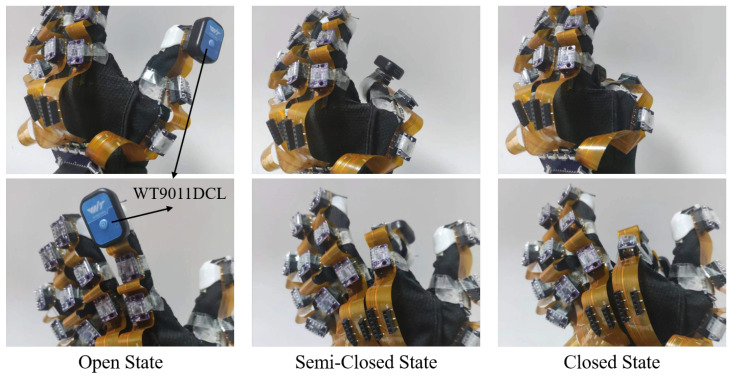
Open state, semi-closed state, and closed states for static tests.

**Figure 10 biomimetics-08-00083-f010:**
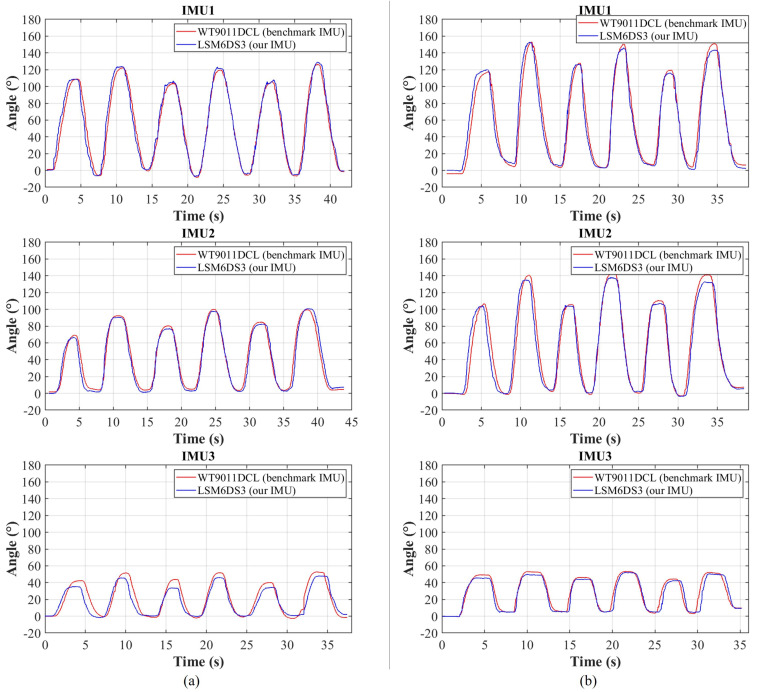
Comparison of benchmark and our IMU results for dynamic tests for (**a**) the thumb and (**b**) index finger.

**Figure 11 biomimetics-08-00083-f011:**
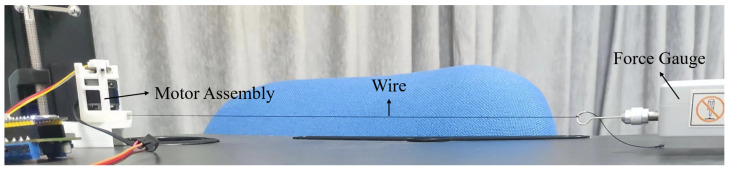
Experimental setup to determine the relationship between the current limit and feedback force.

**Figure 12 biomimetics-08-00083-f012:**
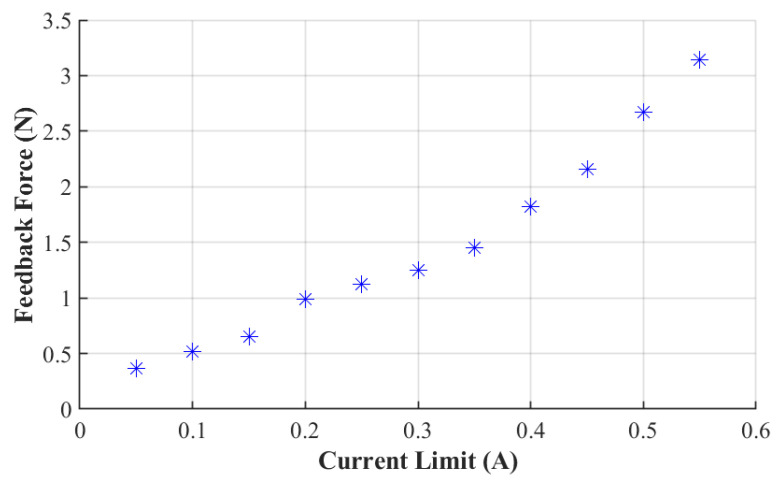
Resulting feedback force versus current limit. Each “*” represents one data point obtained from the test shown in Figure 11.

**Figure 13 biomimetics-08-00083-f013:**
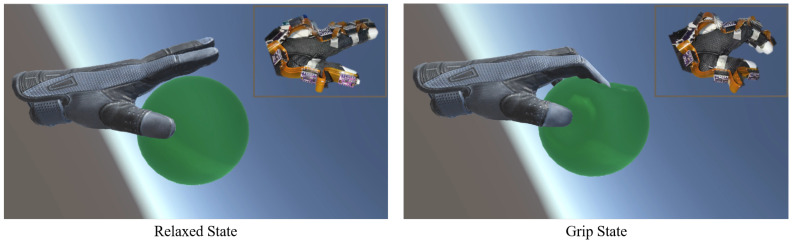
Real-time VR scene application.

**Table 1 biomimetics-08-00083-t001:** The mean absolute error (MAE) of the static tests for the index finger IMU1 during fifteen minutes.

	Open State	Semi-Closed State	Closed State
MAE (°)	0.3243	1.1090	2.6092

**Table 2 biomimetics-08-00083-t002:** The mean error of the dynamic tests for the thumb and index finger.

	Thumb			Index Finger		
	IMU1	IMU2	IMU3	IMU1	IMU2	IMU3
Mean Error (°)	0.8415	−1.1802	−2.9858	−0.8697	−1.9731	−1.7348

## Data Availability

The data that support the findings of this study are available from the authors upon reasonable request.

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
