# Peer review of "Soft Robotic Glove with Sensing and Force Feedback for Rehabilitation in Virtual Reality"

_biomimetics, 2023, doi:10.3390/biomimetics8010083_

Round 1

Reviewer 1 Report

This manuscript presents a new glove that could provide force feedback for future VR systems. The concept is interesting and the results are promising. I have two minor questions/suggestions:

1) In Figure 3(a), it seems that those wires are tangled. I suggest the authors to modify that figure and clearly show how those wires were bundled and connected to each finger.

2) What is the maximum frequency of this glove? How fast could the system track the finger movement while providing proper force feedback?

Author Response

Response to Reviewers

Manuscript Title: Soft Robotic Glove with Sensing and Force Feedback for Rehabilitation in Virtual Reality

Manuscript ID: biomimetics-2202923

We thank the reviewers for their comments. We have addressed each comment as documented below. Reviewer comments are shown in regular font and the authors’ responses are shown in bold font. Two versions of the manuscript are enclosed, one with “Track Changes” in Latex format and the other with regular fonts in pdf format.

Comments and Suggestions for Authors

This manuscript presents a new glove that could provide force feedback for future VR systems. The concept is interesting and the results are promising. I have two minor questions/suggestions:

We thank the reviewer for your comments and will present our list of changes based on the following detailed comments.

  1. In Figure 3(a), it seems that those wires are tangled. I suggest the authors to modify that figure and clearly show how those wires were bundled and connected to each finger.

Figure 3(a) has been updated to show the five feedback wires in different colors.  To ensure smooth driving, they are arranged in a cross pattern. We have also updated the wire in the upper left of Figure 3(a) to its integrated state, as only one wire sticks out and the wire end is wound on the bobbin. We have also explained the arrangement in the updated manuscript.

2.What is the maximum frequency of this glove? How fast could the system track the finger movement while providing proper force feedback?

The maximum frequency of the system to track the finger movement while providing proper force feedback is about 45 Hz, which is mainly determined by the time to process the IMU data.

In addition to the reviewer’s thoughtful suggestions, we have corrected a few grammar issues throughout the paper, which are shown in the “Track Changes” in the Latex manuscript.

Reviewer 2 Report

SUMMARY:

This manuscript presents a soft robotic glove for VR rehabilitation applications, with IMU sensing and force feedback. The hardware design and control are clearly described, and several validation experiments are presented.

GENERAL CONCEPT:

The manuscript has technical quality and provides a prototype of a hand rehabilitation tool with high potential. Rehabilitation is moving towards new paradigms based on VR and gamification, aiming to increase patients' motivation. Just some minor issues such as glove bulkiness should be addressed (specially the force feedback module with the drive motors). In general, the manuscript is well written and English language and style are appropriate. A very interesting manuscript.

SPECIFIC COMMENTS:

ABSTRACT

---- Line 9: I would allow using directly “FOC” acronym in the abstract. It is only used once, it has not been previously explained and it’s an acronym quite more technical than VR or IMU.  

1.      INTRODUCTION

The background and motivation of the work is well presented, nevertheless, I have some minor comments:

·  ---- Line 33: However, the accuracy of such sensors is generally low due to the shifting of position during bending [12].

I’ve been using several data gloves, such as VMG30 from Virual Realities, that presented the mentioned problem of gauge position shifting. Nevertheless, I used different models of CyberGlove (CyberGlove Systems), and this kind of gloves have each extensiometric gauge located into a sewn area, not allowing the shifting of their position. Maybe, a stronger reason for discarding it would be that this kind of gloves only provide relative position between finger segments (i.e: joint angles), but do not provide position in space of each segment. I guess that for VR applications, generally, it is important knowing hand position in space (detecting if hand palm is upwards or downwards).  Maybe it is not important for the specific application of ball squeezing, but for future VR scenarios I guess it could be.

2. HARDWARE DESIGN

This section provides all the information required to understand all the hardware part of the glove. Figure1 to Figure3 add a lot of clarity to the information provided in the main text and present all the layout in detail. Information is presented gradually, making the text easy to follow.

Only some minor comments: 

·    ---- Line 77 and 88: Better using “posture of finger joints”, “fingers’ posture” or “finger joint angles” instead of “fingers’ shapes” throughout the whole manuscript.

·   ---- Line 87: I would recommend specifying that MCU1 is the MCU part for motion tracking and MCU2 is the part controlling force feedback. Nevertheless, it can be deduced later from the Figure 1, but it would add clarity to the main text. It’s just a suggestion.

3. SOFTWARE CONTROL

This section provides specific technical data for understanding signal treatment and joint angles computing. The flowchart adds a lot of clarity. Sensor calibration, threshold correction and signal filtering are detailed. Just a comment:

·  ---- Line 197 and caption of Figure 6: please, replace “phalangeal” for “interphalangeal”.

4. EXPERIMENTAL VALIDATION AND APPLICATION

This section presents a validation of IMUs data both in static and dynamic conditions, along with empirical results supporting it. The current limit and the force feedback study is also detailed, and an overview of the rehabilitation application developed in Unity is presented. The supplementary video illustrates the accuracy of real-time motion capture and also the use of the glove with the VR application. This section adds the consistency that a glove of this characteristics requires.

5. CONCLUSION

Main strengths of the glove and validations performed are remarked in this section, providing an overview of all the validation experiments performed.

Finally, the future work is outlined. This glove has a lot of potential and several VR scenes could be implemented. I am curious about the algorithms for automatic assessment of the motor impairment level, as the only input of the glove is hand joint kinematics. I guess it will compare for each patient the joint ranges of motion throughout the several rehabilitation sessions, checking if they are gradually increasing.

Reviewer 3 Report

The authors proposed a soft robotic glove with sensing and force feedback for hand rehabilitation in virtual reality. This paper detailed the construction of the glove, the force feedback mofule, the system sensing and control algorithms. The performance of the glove is fully verified by extensive experiments. There are a few minor points that need to be clarified.

1.         The first occurrence of IMUs in the abstract should be explained.

2.         The references in the section introduction should be more comprehensive.

3.         The quality of the figures needs to be further improved.

4.         If the conclusion is described separately, the logic will be clearer.
